# Heart Rate Monitor Instead of Ablation? Atrioventricular Nodal Re-Entrant Tachycardia in a Leisure-Time Triathlete: 6-Year Follow-Up

**DOI:** 10.3390/diagnostics10060391

**Published:** 2020-06-10

**Authors:** Robert Gajda

**Affiliations:** Center for Sports Cardiology at the Gajda-Med Medical Centre in Pułtusk, ul. Piotra Skargi 23/29, 06-100 Pułtusk, Poland; gajda@gajdamed.pl; Tel.: +48-604286030

**Keywords:** AVNRT, endurance training, HRM, triathlon, exertion cardiac arrhythmia, Holter ECG

## Abstract

This study describes a triathlete with effort-provoked atrioventricular nodal re-entrant tachycardia (AVNRT), diagnosed six years ago, who ineffectively controlled his training load via heart-rate monitors (HRM) to avoid tachyarrhythmia. Of the 1800 workouts recorded for 6 years on HRMs, we found 45 tachyarrhythmias, which forced the athlete to stop exercising. In three of them, AVNRT was simultaneously confirmed by a Holter electrocardiogram (ECG). Tachyarrhythmias occurred in different phases (after the 2nd–131st minutes, median: 29th minute) and frequencies (3–8, average: 6.5 times/year), characterized by different heart rates (HR) (150–227 beats per minute (bpm), median: 187 bpm) and duration (10–186, median: 40 s). Tachyarrhythmia appeared both unexpectedly in the initial stages of training as well as quite predictably during prolonged submaximal exercise—but without rigid rules. Tachyarrhythmias during cycling were more intensive (200 vs. 162 bpm, *p* = 0.0004) and occurred later (41 vs. 10 min, *p* = 0.0007) than those during running (only one noticed but not recorded during swimming). We noticed a tendency (*p* = 0.1748) towards the decreasing duration time of tachycardias (2014–2015: 60 s; 2016–2017: 50 s; 2018–later: 37 s). The amateur athlete tolerated the tachycardic episodes quite well and the ECG test and echocardiography were normal. In the studied case, the HRM was a useful diagnostic tool for detecting symptomatic arrhythmia; however, no change in the amount, phase of training, speed, or duration of exercise-stimulated tachyarrhythmia was observed.

## 1. Introduction

Heart-rate monitors (HRMs) are devices designed to control the intensity of training in athletes and are not intended to serve as medical devices to diagnose arrhythmias [1]. However, they can “catch” cardiac arrhythmia during training. While indications of sudden, unexpected heart rate (HR) values on HRMs during training, without clinical symptoms are most often ordinary artifacts, arrhythmias on the HRM which force the athlete to stop training cannot remain undiagnosed [2]. Atrioventricular nodal re-entrant tachycardia (AVNRT) is one of the more common, clinically manifesting arrhythmias in endurance athletes, with and without structural heart disease [3]. Athletes present more frequently with an atypical subform of AVNRT. This is possibly related to cardiac remodeling with dilatation of the cardiac cavities, which leads to changed conduction properties in the septal area [4]. It is dependent on the functional dissociation of atrioventricular nodal conduction over a fast and slow pathway or even over two slow pathways, wherein structural and electrical cardiac remodeling occurs with exercise [5]. The ablation of AVNRT carries the very low, but real risk of inducing conduction tissue damage (0.7%) [6]. Therefore, some athletes decline the procedure. Some choose to stop training altogether, while others increasingly limit the intensity of training. There are no previous long-term observational studies of athletes with heart arrhythmia training with HRMs. We examined a triathlete who, for six years after being diagnosed with exercise-stimulated arrhythmia as AVNRT, consistently participated in amateur triathlon training and competitions, while attempting to limit the number of tachycardic events using HRMs.

## 2. Materials and Methods

### 2.1. Sports Biography 

A triathlete, 47 years old, 191 cm tall, weighing 91 kg (body mass index, 24.94), performs a free profession by working mentally, has regularly trained as an amateur from 6 years of age, almost every day, treating sports as a factor of a healthy lifestyle and a great adventure. 

He was not focused on sports competition, and in the relatively few triathlons or cycling competitions he participated in, he finished more or less in the middle of the competitors, advancing in age groups through the years (currently Master-45). For 8 years of training, he felt violent “heart palpitations”, confirmed by high HR values on an HRM, often exceeding 200 bpm, which forced him to stop training until they subsided. Six years ago, he presented to the Center for Sports Cardiology. He reported sudden changes in HR during exercise, from typical for a given phase of training to exceeding the maximum HR. One of the arrhythmias was “caught” on his HRM during cycling training, in which the athlete was also wearing a Holter electrocardiogram (ECG). The arrhythmia registered simultaneously at the 68th min of effort, and HR increased from 167 to 227 bpm, preventing the athlete from continuing (Table A1 pos. 7). The maximum HR of the athlete described, determined empirically (during tests and competitions without arrhythmia, recorded on HRM) was 184 bpm for 6 years and did not change during the observation period. Both devices, i.e., HRM and Holter ECG, noted identical changes in HR values during this symptomatic tachycardia, described as AVNRT in the article by Gajda et al. [7]. Later, AVNRT was recorded on the Holter ECG twice, identical to HR values indicated on the HRM. Clinical studies confirmed the diagnosis of AVNRT with qualification for ablation. However, the triathlete did not report for the proposed treatment. Instead, he trained for the next 6 years with a HRM, collecting data obtained from it on a computer, scrupulously noting all arrhythmia attacks in a training diary. The athlete described tachyarrhythmic attacks as: completely unexpected: mainly at the beginning of training and more often during the running and “predictable” (high probable). The latter appeared most often during long bicycle training in the phases of submaximal effort (and submaximal HR). Sometimes he was able to provoke them with submaximal, interval, long-term training. The start in the competition was a trigger for arrhythmia, which the athlete associated with emotional tension (large adrenergic component) and maximum effort. However, this was not the rule. In adult life, he spent well over 4000 h training and competing in endurance sports (10 years counted), 5–6 times per week. His most preferred triathlon discipline is cycling, and his least is running. He occasionally abstained from physical activity (e.g., during travels). He never got sick and he abstains from alcohol and smoking. During the observation period, he experienced only one arrhythmic attack while swimming, without hemodynamic effects, threatening him with an inability to continue swimming. He never lost consciousness or experienced an arrhythmia while at rest. He did not take any medications (e.g., beta-blockers) that, in the past, occasionally taken gave “discomfort” resulting in abstentions of training. A schematic sport curriculum vitae is presented in Table 1.

### 2.2. Methods

#### 2.2.1. Study Protocol

In the triathlete, whom we diagnosed with AVNRT six years ago, we analyzed about 1800 training sessions recorded on HRMs during the following 6-year period, some compared with the competitor’s notes (training diary). We further analyzed only those trainings in which there were unexpected increases in HR values during training combined with clinical symptoms in the form of a sudden decrease in physical fitness. Sudden “bursts” of HR in training recorded exclusively on HRMs, of which there were about 100 in the analyzed period, without any clinical symptoms, were treated as artifacts, in accordance with previous studies [7]. Finally, we performed 3 simultaneous tests: exercise stress tests, Holter ECGs, and HR measurements using an HRM, to provoke and evaluate exercise arrhythmias and assess the correctness of the HRM indications (Figure 1). We performed ECG and echocardiography directly preceding the exercise stress test. We continued the Holter ECG examination the next day during field training in order to assess HRM indications and record any HR disorders in natural conditions. 

#### 2.2.2. Electrocardiogram (ECG) Tests

Standard 12-lead ECG was performed using a BTL Flexi 12 ECG device (BTL Industries Ltd., Hertfordshire, UK).

#### 2.2.3. Transthoracic Echocardiography

The patient underwent complete transthoracic echocardiographic examination using a GE Medical System Vivid 7 with a 2.5-MHz transducer (GE Medical Systems Information Technologies, Inc., Wauwatosa, WI, USA). M-mode, two-dimensional (2D) imaging, and Doppler techniques were used. The left ventricular (LV) end-systolic and LV end-diastolic volumes and the interventricular septal diastolic and posterior wall thickness diameters were measured. The LV systolic function was evaluated using the LV ejection fraction (LVEF) and longitudinal strain (global longitudinal strain (GLS).

LV diastolic function was evaluated using mitral inflow velocities and tissue Doppler imaging (TDI) values. The transmitral early diastolic (E-wave) velocity and atrial (A-wave) velocity were measured and the E/A ratio was calculated. Early diastolic velocity (e’) was measured in addition to E/e’ ratio.

The right ventricular end-diastolic diameter from the parasternal long-axis view and the tricuspid lateral annular systolic velocity wave (S’RV) were measured using TDI. The left atrial volume index was calculated using the body surface area.

#### 2.2.4. Holter ECG

A 24-h Holter ECG monitoring was performed with the Holter ECG Lifecard CF apparatus and software version: Cardionavigator Plus Impresario 3.07.0158. (Reynolds Medical, Paris, France). 

#### 2.2.5. Exercise Stress Test

A treadmill exercise stress test was performed according to the individually modified protocol (by adjusting the speed and incline of the treadmill to lengthen the submaximal exercise phase) using the set: EKG apparatus BTL Flexi 12 ECG (BTL Industries Ltd., Hertfordshire, UK), treadmill BTL-770M (BTL Industries Ltd., Hertfordshire, UK). Software: BTL CardioPoint 6.1.7601.24545 (UK).

#### 2.2.6. Heart-Rate Monitor (HRM) Analyses

HR measurements were obtained over the 6-year period via the following HRMs: Polar Vantage V (POLAR Electro, Kempele, Finland), Polar V800 HR monitors (POLAR Electro, Kempele, Finland), and Forerunner 910 XT GPS, (Garmin^®^ Ltd., Southampton, UK). All HRMs were tested and compared with Holter ECG measurements. The results were analyzed by a cardiologist with extensive experience.

#### 2.2.7. Statistical Analysis

Normal distributions were analyzed using the Shapiro–Wilk test. For variables without a normal distribution, the significance of differences was measured using the Wilcoxon test (by comparing two groups: type of activity; type of tachycardia event) and the Friedman analysis of variance (ANOVA) (by three groups: period). Any significance in changing tachycardia frequencies was established based on trend analysis. All the statistical calculations were performed using the STATISTICA 12 package (StatSoft, Tulsa, OK, USA). The significance level was set at *p* < 0.05.

#### 2.2.8. Ethical Approval

This case report was approved by the ethical review board of the Bioethics Committee of the Healthy Life Style Foundation in Pułtusk (EC 5/2019/medicine/sports, approval date: 29 June 2019). The runner provided his written informed consent to participate in the analysis and for his data to be published.

## 3. Results

### 3.1. HRM Data Analysis

Only training sessions in which unexpected increases in HR values recorded on a HRM, resulting in a sudden decrease in physical capacity preventing the continuation of training, were analyzed. Training sessions, for which the quality of data were doubtful, were excluded from the analysis. Taking into consideration over 6 years of training, statistically significant differences in the frequency of tachycardia (*p* = 0.6068), their intensity (*p* = 0.2657) or duration (*p* = 0.1748) were not observed, wherein cycling and running training were analyzed together. Nevertheless, there was a statistically insignificant tendency towards decreasing durations of tachycardia (*p* = 0.1748) (Table 2, Figure 2). Analyzing cycling and running training separately, we observed that the intensity of tachycardia i.e., the rate of ventricular rhythm recognized by HRM as HR, was significantly lower during running activities (*p* = 0.0004) and occurred later (*p* = 0.0007). The rate of tachyarrhythmia (HR) on HRM as well as its duration during cycling training was 165–227 bpm (median: 200 bpm) / 15–186 s (median: 45 s), whereas during running training, it was 150–202 bpm (median: 162 bpm)/10–170 s (median: 39 s) (Table 2). No other relevant observations have occurred. On three different occasions, tachycardia occurred twice during the same training session (Table A1, position 22a, 28a, 30a); during one training session, the athlete experienced tachycardia three times (Table A1, position 25, 25a, 25b). The episodes of repeated tachycardia were characterized by higher intensity and decreased duration compared to the non-repeated tachycardia, but these characteristics were not statistically significant (respectively *p* = 0.2249 and 0.1380, Table 2). Only three tachycardias (Table A1: no 7, 19, 36, Table 3: no 1, 3, 6) were simultaneously recorded on both the HRM and Holter EEG, wherein AVNRT was confirmed. All of the other episodes of analyzed tachycardia occurred during training sessions, when the participant was not wearing the Holter ECG device. The amateur-triathlete experienced only one tachycardia during swimming training, without good confirmation on HRM. Full data from heart rate monitors obtained during 6 years of observation are in Table A1.

### 3.2. ECG Tests

During the rest ECG, we observed sinus rhythm 63/min. ECG recording normal. (Figure 3).

### 3.3. Echocardiography

All the evaluated parameters remained within the normal ranges (Table 3).

### 3.4. Exercise Stress Test

The athlete underwent a treadmill exercise stress test, according to an individual protocol, with the additional goals of provoking an arrhythmia and evaluating the associated HRM indications. The effort ended due to fatigue at the 44th min. The maximum heart rate of 182/min was reached in 35 min. At the same time, the highest metabolic equivalent of task (MET) load of 15.9 was achieved. During the modified test (3× longer when usual), several attempts of interval maximum efforts were made to provoke arrhythmia without success. The athlete exhibited a correct pressure response to effort, without chest pain or ST-segment changes in ECG. Conclusion: negative exercise stress test.

### 3.5. Heart-Rate Monitor Tests

We tested the correctness of the indications of all 3 HRMs (Polar Vantage V, Polar V800 HR monitors and Forerunner 910 XT GPS) used over the 6-year period several times. Additionally, we simultaneously tested the Polar Vantage V HRM while the athlete was wearing a Holter ECG during an exercise stress test (Figure 1 and Figure 4, Table 4 No. 8). In 2014, we checked the correctness of the HRM then used (Table 4, No. 1) simultaneously with the Holter ECG, when AVNRT was first detected [7]. During the tests, all three HRMs showed the same maximum and average HR values indicated by the Holter ECG. In total, AVNRT on the Holter ECG was recorded 3 times with the same HR as the HRMs’ indications. Table 4 indicates the detailed values obtained during HRM tests versus the Holter ECG, showing the highest HR values on HRM and HR values shown simultaneously by the Holter ECG.

## 4. Discussion

### 4.1. Discussion of the Results

We analyzed the triathlete’s training, recorded on HRMs for 6 years, with recognized effort-stimulated AVNRT. It is a rare and specific situation when an athlete, using HRM, tries to limit the number of symptomatic tachyarrhythmias. During the observation period, we did not notice significant changes: frequency, intensity (speed) and their duration.

Sports HRMs, in which HR is one of several assessed values, are designed to control the intensity of training [8]. Eight years ago, the triathlete described in this study noticed sudden increases in HR on his HRM, which forced him to stop training. Finally, 6 years ago, tachyarrhythmia was simultaneously recorded on the HRM and Holter ECG, and confirmed at the Cardiology Clinic as AVNRT. The analysis of approximately 1800 training sessions recorded on HRM over the past 6 years has allowed us to segregate and analyze 45 tachyarrhythmias, without artifacts, which were also noted in his training diary. Their analysis indicates no significant changes in the frequency of arrhythmias, their speed, length, and the phase of training in which they occurred (Table 2, Figure 2). It is not clear why there is a significant difference in maximal HR during arrhythmia generated during running in relation to cycling. The onset of arrhythmia was also significantly earlier during running (Table 2). The triathlete definitely preferred cycling, which could explain his better adaptation to this type of effort and, thus, the delayed arrhythmia generation. However, this is only speculation. It can be speculated that the fact that there is no arrhythmia during swimming (a one-time episode felt clinically without documentation in the form of a recording from HRMs due to a technical error in recording) is due to the much lower muscular involvement associated with working mainly the upper body during swimming and the decreased stress on the circulatory system. There was an average of 12 AVNRT seizures recorded in the training notebook per year, but the HRM device was only able to reproduce an average of 7/year as “technically well documented”. The athlete, refusing ablation, hoped that with time the arrhythmias would subside, slow down, or shorten in duration. Apart from the slight tendency to shorten in duration, this did not happen. It also seems that attempts to reduce the number of seizures by controlling HR and well-being were ineffective, as evidenced by the primary arrhythmias recorded, as well as the three secondary and 1 tertiary tachycardic episodes (Table 2, Table A1). Modified training, aimed at provoking arrhythmias (long-term and submaximal), turned out to be ineffective. No other rules or trends other than those listed were observed. No other factors were found that triggered the, unexpected and anticipated” arrhythmias by analyzing them all together (e.g., the type of the beginning of training, special warm-up, breaks in training, fast start, etc.) Listed by the athlete as provocative for arrhythmias: long, submaximal, interval training, or start in competitions, although often, they were not always triggers of arrhythmias. Judged together with the “unexpected” in the initial phase, they did not allow drawing any additional conclusions or observations. Tachyarrhythmia, each time it occurred, was a very unpleasant feeling for the athlete, which forced him to stop training until it subsided. The echocardiographic examination and exercise stress test did not show any abnormalities. The ECG markers related to sudden cardiac death in the examined athlete, such as P-wave (duration, interatrial block, and deep terminal negativity of the P wave in V1), prolonged QT and Tpeak-Tend intervals, QRS duration and fragmentation, bundle branch block, ST segment depression and elevation, T waves (inverted, T wave axes), premature ventricular contractions, and ECG hypertrophy criteria are missing [9].

### 4.2. Atrioventricular Nodal Re-Entrant Tachycardia (AVNRT) in Athletes

The development of an athlete’s heart is recognized as a risk factor for atrial arrhythmias with AVNRT. Prolonged participation in exercise (moderate to intensive sports for ≥3 h per week for ≥5 years) results in structural and electrical cardiac remodeling. Athletes with AVNRT present more frequently with an atypical forms of AVNRT than non-athletes [4]. Exercise can be implicated as an independent causal factor in the development of AVNRT and is more often involved in the development of atypical subform [4]. Dilatation of the right atrium, as seen in the athletic population, may lead to further stretching of the posteroseptal area, facilitating anisotropy. This in turn forms the basis of dual atrioventricular (AV) nodal conduction pathways and the circuit of AVNRT [10]. In the case of the athlete under study, the right atrium in the echocardiography was normal, which, however, does not exclude changes that could be seen during magnetic resonance imaging (MRI). The effect of the endurance exercise on the atria is more pronounced in men than in women [11]. This could be attributed to the fact that men tend to exercise more intensively than women [12]. The examined leisure-time athlete practiced various physical activities since childhood and spent over 10 years training for endurance activity, practicing almost every day. His goal was never competition and he treated sport as a “healthy lifestyle”. However, cumulatively, the amount of activity is impressive and should evoke the adaptation and development of the characteristic “athlete’s heart” which, were not observed in ECG and echocardiography.

### 4.3. The Role of HRMs in Recognizing the Type of Arrhythmia in Athletes

HRMs are devices designed to control training via suitable stimuli, including HR assessments [13]. While constantly measuring HR, they may accidentally “catch” arrhythmias [14]. The use of technology has increased tremendously, by means of more reliable, smaller, more accessible, and, especially, more user-friendly devices, which provide a wider range of features, and promote significant benefits for the population and for health professionals [15]. However, the type of arrhythmia is usually impossible to recognize because of the inability of HRMs to interpret a standard ECG, i.e., the inability to interpret P-waves and QRS complexes [16]. There are smartphone applications on the market that register an incomplete ECG recording (i.e., lead I), with the option of recognizing atrial fibrillation and other arrhythmias [17]. Unfortunately, despite this paradigm shift, ECG recording during activity, e.g., running, is very difficult and constant monitoring is completely impossible. HRMs with the “heart-rate variability” (HRV) function enable differentiation of regular and irregular arrhythmias and provide more information than standard ones. However, we were unable to tell whether this athlete’s heart rate variability resulted from premature atrial or ventricular contraction or just irregular sinus rhythm [18]. For some time, HR monitoring under water has also been possible, which allows us to control swim training [19]. Unfortunately, this was not the case with the athlete in this study. Despite the indisputable value for training, many outstanding athletes do not use or stop using HRM, citing an unjustified dispersion of concentration associated with artifacts in the form of false indications of high HR values, which discourages athletes from controlling HR, especially during competitions [20]. However sometimes, HRM can save lives, not only allowing the athlete to determine his location (especially helpful when training in unknown terrain) using Global Positioning System (GPS) functions, but also analyzing HR as a potential cause of loss of consciousness during training [21]. Research conducted by Gajda et al. indicates that the vast majority of sudden unexpected indications of high HR on HRM in asymptomatic patients are mere artifacts, stating that HRMs are not good tools for diagnosing asymptomatic arrhythmias [7]. A single large study confirmed the usefulness of HRMs and smartphone applications used by athletes to recognize symptomatic arrhythmia [22], which was later confirmed only in small groups [23]. These tools mainly confirm arrhythmia (exactly the rate of ventricular rhythm during tachycardia) without the ability to accurately identify its type. This is not resolved by the HRV function mentioned above, which determines the time between R waves (R to R) and thus detects irregularities, indirectly indicative of arrhythmias. Unfortunately, this function in most HRMs works only at rest and cannot be used during training. Two of the three HRMs used in this study (Polar Vantage V and Polar V800 HR) had this function. Only training with the Holter ECG can determine the diagnosis of AVNRT with high probability. For this reason, it is uncertain whether this athlete’s arrhythmias were AVNRT, atrial flutter, atrial fibrillation, or another supraventricular or ventricular tachyarrhythmia. Without an electrophysiological examination of the provoked arrhythmia, AVNRT can be mistaken for a different condition (e.g., posteroseptal catecholaminergic atrial tachycardia) [5]. These limitations seem to negate HRMs in the diagnosis of symptomatic arrhythmia. However, they are not medical devices and are not used to diagnose medical conditions. On the other hand, when arrhythmias are accompanied by clinical symptoms, they force athletes to consult a doctor. The examined triathlete, diagnosed with AVNRT in the Cardiology Clinic in 2014, refused the proposed ablation, and continued amateur training, hoping that the symptoms would subside or abate. None of these expectations came true. He never clinically registered or reported a tachyarrhythmia seizure at rest on HRMs. However, it is uncertain whether there was an asymptomatic arrhythmia during the resting periods, which is not uncommon for many athletes [24]. The essence of the observations is that the practical use of 3 types of HRMs (Polar Vantage V, Polar V800 HR, Forerunner 910 XT GPS) recognized symptomatic tachyarrhythmia; thus, HRMs, originally intended for healthy athletes with normal HRs, may act as a “paramedical” device which may have practical, medical significance in cases of symptomatic arrhythmia. 

### 4.4. Limitation

The main limitation of this study is the uncertainty regarding the type of symptomatic arrhythmia that was recorded by the HRMs. Each episode could have been AVNRT, as recognized in 2014, or it could have been any other supraventricular or ventricular tachyarrhythmia. Another limitation is the fact that there is no complete assessment of the tendency of changes in speed, duration, and frequency for this arrhythmia, because about 25% of the clinically noted arrhythmias were not recorded by the HRMs for various technical reasons (battery defect in transmitter, poor adhesion transmitter belt, low battery in the receiver, loss of register due to accidental deletion, not switching on HRM). The athlete “felt” arrhythmias on average 12 times/year, of which 7–8/year were recorded. Another limitation of the study is the fact that there were only three training sessions in which arrhythmia were simultaneously recorded on HRM and Holter ECG (AVNRT each time). Despite numerous tests with HRMs and a parallel Holter ECG test, including effort provocation, arrhythmias could not be provoked “on demand” for accurate analysis. Moreover, the athlete has an additional Holter ECG test confirming AVNRT during bicycle training, wherein he did not turn on HRM, which excludes the examination from evaluation. 

### 4.5. Perspectives

It seems that the most favorable situation would be to encourage athletes to undergo electrophysiological diagnostics and conduct the best type of therapy based on its result, which, in the case of confirmation of AVNRT, is ablation. Regardless, during this time, you can encourage the athlete to use the appropriate smartphone application to control training. Strip-generating smartphone products (Kardia Mobile by AliveCor and ECG Check by Cardiac Designs) are more powerful at arrhythmia detection than wearable monitors. In practice, they record an ECG in the form of one or several leads. The fact that you have to stop training at the moment is not a problem because arrhythmia already forces the athlete to pause until it stops (except QardioMD) [17]. A paradigm shift in this matter has already taken place with the appearance of this function (ECG recording). Improvements, such as the possibility of continuous registration at any time and without time limits and above all during training, will ensure success and start a new era in the meaning of these devices. Originally helpful in training, these devices may become diagnostic arrhythmia medical tools essential for the safety of every athlete or person leading a healthy lifestyle. 

## 5. Conclusions

During the 6-year observation period, this athlete did not experience any significant decrease in the number, frequency, or speed of the exercise-stimulated arrhythmias. However, his HRMs acted as a useful diagnostic tool for detecting and documenting symptomatic arrhythmias. Moreover, in the future, HRMs, originally intended for healthy athletes with normal HRs, may act as a “paramedical” device, which may have practical, medical significance in cases of symptomatic arrhythmia. 

## Figures and Tables

**Figure 1 diagnostics-10-00391-f001:**
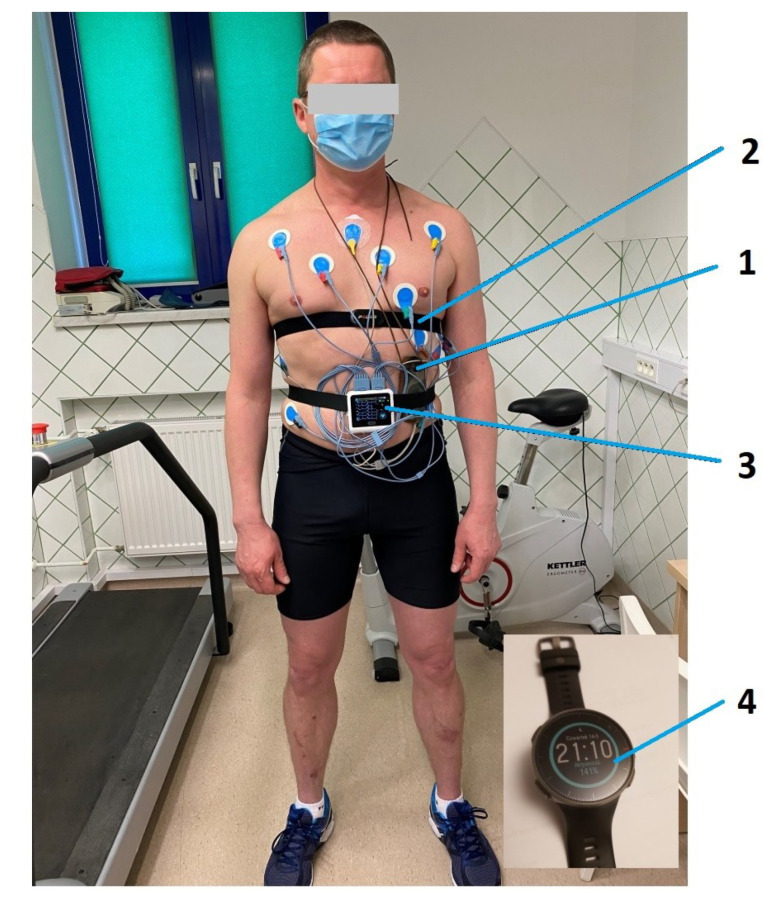
Prepared athlete before exercise stress test on treadmill along with Holter electrocardiogram (ECG) and heart-rate monitor (HRM). 1. Holter ECG, 2. HRM strap, 3. ECG device—from the treadmill exercise stress test set, 4. HRM Polar Vantage V.

**Figure 2 diagnostics-10-00391-f002:**
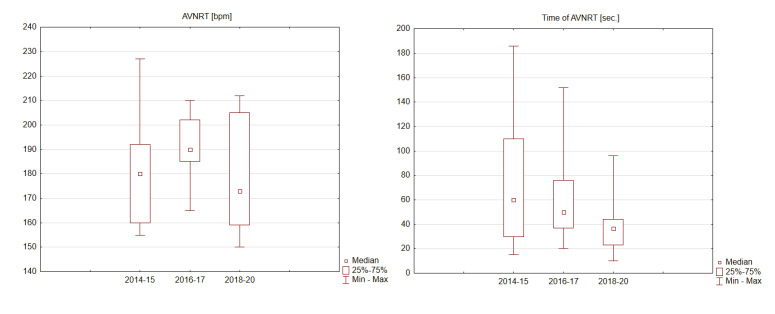
Intensity and time of atrioventricular nodal re-entrant tachycardia (AVNRT) in 2014–2020 of analyzed patient.

**Figure 3 diagnostics-10-00391-f003:**
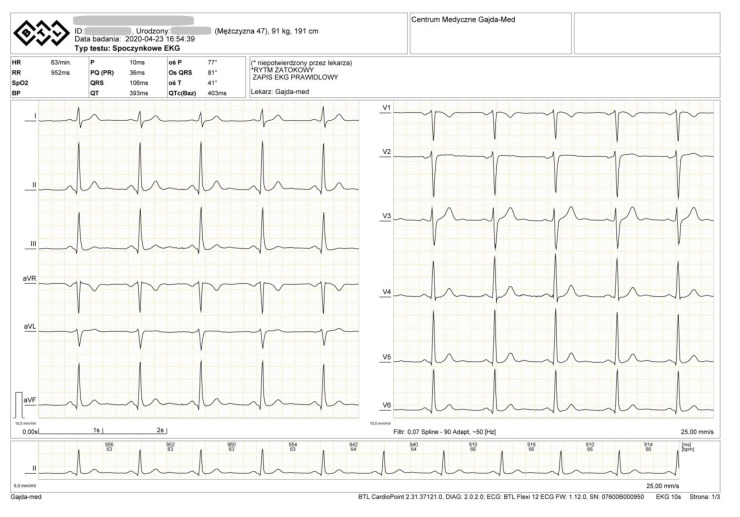
ECG of the examined athlete.

**Figure 4 diagnostics-10-00391-f004:**
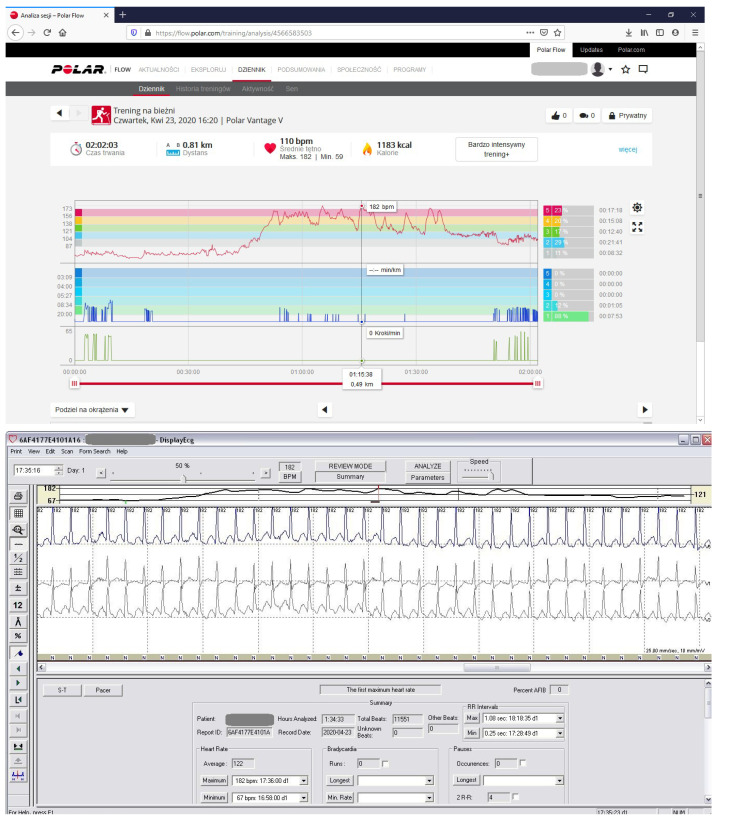
HRM control by comparing heart rate (HR) on HRM and at the same time in Holter ECG. The diagram with HRM indicates max. HR 182/min. At the same time, the Holter ECG indicates a rhythm of 182/min. The test was performed during a modified effort test on a treadmill. (number 8 in Table 4).

**Table 1 diagnostics-10-00391-t001:** Involvement in individual sports disciplines of the examined athlete—‘sports biography’.

Age (Years)	Discipline	Training—Number of Hours/Week	Number of Years on Training	Hours of Training in Discipline	X- Endurance Sports—Adult Age—Hours at Training
6–8	swimming	4	3	600	
8–11	karate kyokushinkai	4	4	800	
12–14	judo	4	3	600	
15–17	swimming	2	3	300	
15–19	volleyball	7	5	1750	
20–26	kickboxing	6	7	2100	
26–27	sambo sports	6	2	600	
27–32	squash	1	6	300	
27–45	cycling	2	18	1800	X
35–45	swimming	2	10	1000	X
35–45	running	2	10	1000	X
35–45	starts in competitions or tests triathlon/running/cycling/swimming	0.5	10	250	X
17–47	skiing, sailing, waking, other	occasionally	30		
Sport-life in total: 6 to 47 = 41 years	all disciplines	5	40	11,100 (about)	X—about 4050 h in total, on average 6 times/week/10 years

**Table 2 diagnostics-10-00391-t002:** Main characteristics of tachycardia events in the analyzed athlete.

Characteristic		Parameters	Start of Tachycardia [min]	HR at Beginning of Tachycardia [bpm]	HR at the end of Tachycardia [bpm]	AVNRT [bpm]	Time of AVNRT [s]
**Type of activity**	All N = 40	Median	29	127	119	187	40
		QD	23.7	11.5	8.8	19.0	21.8
		Min-Max	2–131	86–167	90–156	150–227	10–186
	Cycling N = 20	Median	41	129	117	200	45
		QD	23.7	14.3	18.5	10.0	17.5
		Min-Max	14–131	86–167	90–156	165–227	15–186
	Running N = 20	Median	10	126	119	162	39
		QD	18.0	8.0	6.3	9.5	22.5
		Min-Max	2–64	100–153	98–132	150–202	10–170
	*p*-value (Cyc. vs. Run)		0.0007	0.9679	0.681323	0.0004	0.6149
**Analyzed years with tachycardia**	2014–15 N = 15	Median	23	125	120	180	60
		QD	12.6	13.5	8.5	16.0	40.0
		Min-Max	4–68	90–167	90–156	155–227	15–186
	2016–17 N = 11	Median	33	130	115	190	50
		QD	18.9	13.5	14.0	8.5	19.5
		Min-Max	14–110	86–153	93–139	165– 210	20–152
	2018–IV 20 N = 14	Median	32	127	122	173	37
		QD	34.4	8.5	7.5	23.0	10.5
		Min-Max	2–131	100–147	98–140	150–212	10–96
	*p*-value (comparing periods)		0.1778	0.1778	0.1593	0.2657	0.1748
**Features of tachycardia**	Primary N = 40	Median	29	127	119	187	40
		QD	23.7	11.5	8.8	19.0	21.8
		Min-Max	2–131	86–167	90–156	150–227	10–186
	Repeated N = 5	Median	180	114	109	201	23
		QD	82.4	5.0	2.0	11.0	13.5
		Min-Max	15–191	93–130	79–118	158–217	7–56
	*p*-value (primary vs. repeated)		0.2249	0.2249	0.1056	0.2249	0.1380

Legend: Cyc. vs. Run—Cycling versus Running, QD—Quartile Deviation.

**Table 3 diagnostics-10-00391-t003:** Heart systolic and diastolic function in echocardiographic parameters.

Parameters	Units (Normal Values)	Result
Left ventricle end-diastolic diameter volume	mL (106 ± 22)	111
Left ventricle end-systolic diameter volume	mL (41 ± 10)	35
Ejection fraction 2D (%) bi-plane	% (62 ± 5)	65
Global longitudinal strain	% (−20)	20.4
Interventricular septum diameter	mm (6–10)	10
Posterior wall diastolic diameter	mm (6–10)	10
Right ventricular end-diastolic diameter	mm (20–30)	30
S’ right ventricle	cm/s (14.1 ± 2.3)	16
Left atrium	mm (30–40)	36
Left atrial volume index	mL/m^2^ (16–34)	30.0
Right atrial area	cm^2^ 16 ± 5	14.4
Mitral valve E-wave	cm/s 73 ± 19	80
Mitral valve A-wave	cm/s (69 ± 17)	50
E’ lateral	cm/s (>10)	20
E’ septal	cm/s (>7)	12
E/e’ lateral	ratio(<15)	4.0
E/e’ septal	ratio (<13)	6.6

**Table 4 diagnostics-10-00391-t004:** Tests for the correctness of the HRMs’ indications used by the athlete versus Holter ECG.

Measurement Number	Polar Vantage V (Maximum HR Recorded during Training) (bpm)	V800 HR Monitors (Maximum HR Recorded during Training) (bpm)	Forerunner 910 XT GPS Maximum HR Recorded during Training (bpm)	Holter ECG Maximum HR Recorded during Training(bpm)	Difference in bpm Between Devices
1			227	226 AVNRT	1(App.1 No 7)
2			179	179 SHR	0
3		202		203 AVNRT	1(App.1 No 19)
4		178		178 SHR	0
5		177		178 SHR	1
6	205			206 AVNRT	1(App.1 No 36)
7	174			173 SHR	1
8	182			182 SHR	0 (Figure 1, Figure 4)

Legend: SHR—sinus heart rate, AVNRT—atrioventricular nodal re-entrant tachycardia.

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
