# Peer review of "Heart Rate Monitor Instead of Ablation? Atrioventricular Nodal Re-Entrant Tachycardia in a Leisure-Time Triathlete: 6-Year Follow-Up"

_diagnostics, 2020, doi:10.3390/diagnostics10060391_

Round 1

Reviewer 1 Report

The article by Robert Gajda nicely describes the use of heart rate monitors in a triathlete with effort-provoked atrioventricular nodal re-entrant tachycardia clinically followed/up for six years.

The case is described in detail, and the topic is highly relevant. In this athlete, the use of heart rate monitors acted as a diagnostic tool for documenting arrhythmias. As correctly underlined, these tools were originally intended for healthy athletes but may potentially become paramedical devices shortly. We thus need to collect data on these first "diagnostic" experiences. 

The sole limitation I can point out is that we are not sure about the exact type of arrhythmia occurring in each episode: it might be atrioventricular nodal re-entrant tachycardia, or it could have been any other supraventricular or ventricular tachyarrhythmia. However, the uncertainty regarding the type of arrhythmia that was recorded by the heart rate monitors is a limitation of the currently commercially available heart rate monitors and should not be considered a limitation of this study. Moreover, this point was already underlined by the author in the limitation section.

Author Response

Dear Reviewer,
Thank you for the time devoted to my article, its thorough analysis and acceptance in its current form
greetings
Robert Gajda 

Reviewer 2 Report

The present paper aims to present the case of a triathlete diagnosed with atrioventricular nodal re-entrant tachycardia, who tried to control his training load via heart rate monitors for six years.

A few revisions are needed, as follows:

Results: How do you assess intensity of tachycardia? Please clarify!

Results, line 159-160: You state: “The scope of intensity and duration of tachycardia during cycling training was…” Please rephrase because it is difficult to understand!

Table 2 is difficult to understand. Please provide more explanations!

Discussion should start with your results, emphasizing the particularity of your case and what you noticed during the 6 years follow-up. After that you can try to explain what you noticed.

Please emphasize that ECG markers related to sudden cardiac death are missing in your patient, such as “P wave (duration, interatrial block, and deep terminal negativity of the P wave in V1), prolonged QT and Tpeak-Tend intervals, QRS duration and fragmentation, bundle branch block, ST segment depression and elevation, T waves (inverted, T wave axes), premature ventricular contractions, and ECG hypertrophy criteria.” (Mozos I et al. Electrocardiographic Predictors of Cardiovascular Mortality. Dis Markers. 2015;2015:727401. doi: 10.1155/2015/727401).

Author Response

Dear Reviewer,
Thank you for the time spent on careful analysis of my article. I have used all your comments correcting the article accordingly. I attach the corrected text (in change mode). I hope you are satisfied with all the corrections. Thanks to them, work undoubtedly gained value. In case of doubt or the need for further corrections, I remain at your disposal.
Yours sincerely
Robert Gajda
